# Insights into P-Glycoprotein Inhibitors: New Inducers of Immunogenic Cell Death

**DOI:** 10.3390/cells9041033

**Published:** 2020-04-22

**Authors:** Joanna Kopecka, Martina Godel, Silvia Dei, Roberta Giampietro, Dimas Carolina Belisario, Muhlis Akman, Marialessandra Contino, Elisabetta Teodori, Chiara Riganti

**Affiliations:** 1Department of Oncology, University of Torino, via Santena 5/bis, 10126 Torino, Italy; joanna.kopecka@unito.it (J.K.); martina.godel@edu.unito.it (M.G.); dimascarolina.belisario@unito.it (D.C.B.); muhlis.akman@unito.it (M.A.); 2Department of Neurosciences, Psychology, Drug Research and Child Health, Section of Pharmaceutical and Nutriceutical Sciences, University of Firenze, via Ugo Schiff 6, 50019 Sesto Fiorentino, Italy; silvia.dei@unifi.it (S.D.); elisabetta.teodori@unifi.it (E.T.); 3Department of Pharmacy-Pharmaceutical Sciences, University of Bari, via Orabona 4, 70125 Bari, Italy; giampietroroberta@gmail.com (R.G.); marialessandra.contino@uniba.it (M.C.)

**Keywords:** doxorubicin resistance, P-glycoprotein, calreticulin, triple negative breast cancer

## Abstract

Doxorubicin is a strong inducer of immunogenic cell death (ICD), but it is ineffective in P-glycoprotein (Pgp)-expressing cells. Indeed, Pgp effluxes doxorubicin and impairs the immunesensitizing functions of calreticulin (CRT), an “eat-me” signal mediating ICD. It is unknown if classical Pgp inhibitors, designed to reverse chemoresistance, may restore ICD. We addressed this question by using Pgp-expressing cancer cells, treated with Tariquidar, a clinically approved Pgp inhibitor, and *R*-3 compound, a *N*,*N*-bis(alkanol)amine aryl ester derivative with the same potency of Tariquidar as Pgp inhibitor. In Pgp-expressing/doxorubicin-resistant cells, Tariquidar and *R*-3 increased doxorubicin accumulation and toxicity, reduced Pgp activity, and increased CRT translocation and ATP and HMGB1 release. Unexpectedly, only *R*-3 promoted phagocytosis by dendritic cells and activation of antitumor CD8^+^T-lymphocytes. Although Tariquidar did not alter the amount of Pgp present on cell surface, *R*-3 promoted Pgp internalization and ubiquitination, disrupting its interaction with CRT. Pgp knock-out restores doxorubicin-induced ICD in MDA-MB-231/DX cells that recapitulated the phenotype of *R*-3-treated cells. Our work demonstrates that plasma membrane-associated Pgp prevents a complete ICD notwithstanding the release of ATP and HMGB1, and the exposure of CRT. Pharmacological compounds reducing Pgp activity and amount may act as promising chemo- and immunesensitizing agents.

## 1. Introduction

The ability of cancer cells to escape from the immune system and the cytotoxicity of chemotherapy are two reasons for bad prognosis in oncological patients. To induce an immune-mediated killing, dying cancer cells must produce several damage-associated molecular pattern (DAMP) messengers that cause cancer cells to be recognized by immune cells and trigger the so-called immunogenic cell death (ICD) [1,2]. Both surface and soluble molecules are required to induce ICD in response to chemotherapeutic drugs such as anthracyclines, oxaliplatin, taxanes, alkylating agents, and proteasome inhibitors [2,3,4]. The exposure of calreticulin (CRT) on cell surface and the release of ATP and High Mobility Group Box 1 (HMGB1) protein are potent signals that engage dendritic cells (DCs), stimulate phagocytosis and antigen processing, and expand cytotoxic CD8^+^T-lymphocytes (CTLs) with antitumor activity [2,5]. CRT, a chaperon protein usually residing within endoplasmic reticulum (ER), translocates to the plasmamembrane, bound to the ERp57 protein. By engaging the LDL receptor-related protein 1/CD91 receptor on DCs, the dimer CRT/ERp57 initiates the phagocytosis of dying cancer cells [6,7]. Soluble ATP binds the purinergic receptor P2X7 on DCs; this step is followed by the activation of NOD-like receptor family, pyrin domain containing-3 protein (NLRP3)-dependent caspase-1 complex. This complex, also known as the inflammasome, increases the production of IL-1β, facilitating the recruitment of CTLs producing IFN-γ [8]. ATP is necessary to induce ICD, as demonstrated by the abrogation of chemotherapy-induced ICD in the presence of the activated ecto-ATPases CD39 [9] and CD73 [10]. HMGB1, together with the surface heat shock protein 90 (HSP90), acts as a ligand of Toll-like receptor-4 (TLR4): this interaction stimulates the antigens processing by DCs, which is the premise to expand CTLs directed against cancer cells [11]. Overall, the cooperation between CRT-, ATP-, and HMGB1-dependent signals is crucial to promote DC homing and recruitment, antigens processing, and antitumor CTLs engagement [12].

Several factors may limit the efficacy of ICD, such as the tumor mutational burden; the tumor microenvironment containing specific populations, e.g., cancer-associated fibroblasts (CAFs), M2-tumor-associated macrophages (TAMs), myeloid-derived suppressor cells (MDSCs), and T-regulatory cells (Tregs), inhibiting the expansion of CTLs; and the chemotherapy protocols based on few administrations of the maximum tolerated dose instead of metronomic protocols [13]. 

In addition, another factor impairing ICD is the intrinsic or acquired resistance to chemotherapy. Doxorubicin, a topoisomerase IIα inhibitor, is an excellent inducer of ICD [14]. However, we demonstrated that cells resistant to necro-apoptotic death induced by doxorubicin are also resistant to ICD [15,16,17]. One of the main mechanisms involved in the resistance to doxorubicin is the presence on the cell surface of efflux proteins belonging to the ATP Binding Cassette (ABC) transporters, such as ATP Binding Cassette transporter B1 or P-glycoprotein (ABCB1/Pgp), ATP Binding Cassette transporter C1 or multidrug resistance-related protein 1 (ABCC1/MRP1), and ATP Binding Cassette transporter G2 or breast cancer resistance-related protein (ABCG2/BCRP) [18]. By reducing the intracellular concentration of doxorubicin, these proteins limit all of doxorubicin’s death mechanisms, including ICD [15]. Furthermore, we demonstrated that Pgp, which is synthesized within the ER, is transported to the plasma membrane, following the same mechanism of CRT: here, Pgp co-localizes with CRT and prevents cell phagocytosis by DCs [19]. In the last decades, three generations of Pgp/interacting compounds, including substrates, inhibitors, or modulators [20], have been developed with the aim of reversing chemoresistance in Pgp-expressing tumors [21]. Most of them failed because of heavy side effects, unexpected drug–drug interactions with chemotherapeutic agents, or poor pharmacokinetic profiles [20]. Tariquidar is one of the latest compounds approved for clinical trials: it is tolerable and does not impair the pharmacokinetics of chemotherapeutic drugs; unfortunately, the real efficacy in terms of anticancer effects is doubtful and it has not been approved for therapy [22]. Until now, there are no studies investigating if Pgp inhibitors can also rescue the ICD induced by chemotherapeutic Pgp substrates like doxorubicin.

In the present work, we investigated how Pgp impairs doxorubicin-mediated ICD and if Pgp inhibitors, originally designed as chemosensitizing agents, may be considered as ICD-inducers in combination with doxorubicin. To this aim, we compared Tariquidar and compound *R*-3—the first-in-class compound of a library of *N*,*N*-bis(alkanol)amine aryl ester derivatives recently synthesized by our group [23]—that has an EC_50_ for Pgp activity (73 nM) comparable to Tariquidar (74 nM) [24]. We demonstrated that the reduction of Pgp on cell surface but not the simple inhibition of the transporter activity is required to restore doxorubicin-mediated ICD.

## 2. Materials and Methods

### 2.1. Chemicals

Fetal bovine serum (FBS) and culture medium were acquired from Invitrogen Life Technologies (Carlsbad, CA, USA). Plasticware for cell cultures was from Falcon (Becton Dickinson, Franklin Lakes, NJ, USA). The protein content in cell extracts was assessed with the BCA kit from Sigma-Merck (St. Louis, MO, USA). Electrophoresis reagents were obtained from Bio-Rad Laboratories (Hercules, CA). Doxorubicin and Tariquidar were purchased by Sigma-Merck. *R*-3 synthesis and characterization is reported in [23]. The structures of Tariquidar and *R*-3 are shown in Appendix A.

### 2.2. Cells

Human doxorubicin-sensitive colon cancer HT29 cells, non-small cell lung cancer A549 cells, and triple-negative breast cancer MDA-MB-231 cells were purchased from ATCC (Manassas, VA, USA). The corresponding doxorubicin-resistant variants (HT29/DX, A549/DX, and MDA-MB-231 DX) were selected by culturing parental cells in medium containing progressively increasing doxorubicin dosages [25], and maintained at a final doxorubicin concentration of 200 nM (HT29/DX), 100 nM (A549/DX), and 200 nM (MDA-MB-231/DX). Culture medium was supplemented with 10% *v*/*v* FBS, 1% *v*/*v* penicillin–streptomycin, and 1% *v*/*v*
l-glutamine. All cell lines were authenticated by microsatellite analysis, using the PowerPlex kit (Promega Corporation, Madison, WI, USA; last authentication: November 2019).

### 2.3. Immunoblotting

Cells were lysed in MLB buffer (125 mM Tris-HCl, 750 mM NaCl, 1% *v*/*v* NP40, 10% *v*/*v* glycerol, 50 mM MgCl_2_, 5 mM EDTA, 25 mM NaF, 1 mM NaVO_4_, 10 mg/mL leupeptin, 10 mg/mL pepstatin, 10 mg/mL aprotinin, 1 mM phenylmethylsulphonyl fluoride PMSF, pH 7.5), sonicated and centrifuged at 13,000× *g* for 10 min at 4 °C. Fifty micrograms of proteins was subjected to immunoblotting and probed with the following antibodies: anti-ABCB1/Pgp (C219, Novus Biologicals, Littleton, CO, USA; dilution 1/250), anti-ABCC1/MRP1 (IU2H10, Abcam, Cambridge, UK; dilution 1/100), anti-ABCG2/BCRP (sc-25882, Santa Cruz Biotechnology Inc., Santa Cruz, CA, USA; dilution 1/500), followed by a peroxidase-conjugated secondary antibody. Proteins were detected by enhanced chemiluminescence (Bio-Rad Laboratories). Plasma membrane-associated proteins were evaluated in biotinylation assays, using the Cell Surface Protein Isolation kit (Thermo Fisher Scientific Inc., Rockford, IL, USA) [15], and probed with anti-Pgp and anti-CRT (PA3-900, ABR-Affinity BioReagents Inc., Golden, CO, USA; dilution 1/500) antibodies. Non-biotinylated proteins, i.e., cytosolic proteins, were blotted with the anti-Pgp antibody. Anti-β-tubulin antibody (sc-5274, Santa Cruz Biotechnology Inc., Santa Cruz, CA, USA; dilution 1/1000) was used as control of equal protein loading in cytosolic extracts; anti-pancadherin antibody (CH-19; Santa Cruz Biotechnology Inc., dilution 1/500) was used as control in plasma membrane extracts. In co-immunoprecipitation experiments, 100 μg of plasma membrane-associated proteins were immunoprecipitated with the anti-CRT antibody, using PureProteome protein A and protein G Magnetic Beads (Millipore, Bedford, MA, USA), and then blotted for Pgp. To assess Pgp ubiquitination, 50 μg whole cell lysate was immunoprecipitated with the anti-Pgp antibody, and then probed with an anti-mono/polyubiquitin antibody (FK2, Axxora, Lausanne, Switzerland; dilution 1/1000).

### 2.4. Intracellular Doxorubicin Accumulation and Doxorubicin Kinetic Efflux

The intracellular doxorubicin content and the drug efflux were measured as detailed in [26]. The intracellular doxorubicin concentration was expressed as nanomoles doxorubicin/mg cellular proteins. The efflux of doxorubicin was expressed as the change in the intracellular concentration of the drug/minute (dc/dt) [26]. Km and Vmax parameters were estimated using the Enzfitter software (Biosoft Corporation, Cambridge, UK).

### 2.5. ATPases Activity

Pgp, MRP1, and BCRP were immunoprecipitated from 100 μg of membrane-associated proteins, then the rate of ATP hydrolysis, an index of the catalytic cycle and a necessary step for substrate efflux, was measured spectrophotometrically [27]. In each set of experiments, 0.5 mM Na_3_VO_4_ was included in the reaction mix to measure the Na_3_VO_4_-sensitive rate of ATP hydrolysis. Results were expressed as nmoles hydrolyzed phosphate/mg protein.

### 2.6. Caspase 3 Activity

Cells were lysed in 0.5 mL of lysis buffer (20 mM Hepes/KOH, 10 mM KCl, 1.5 mM MgCl_2_, 1 mM EGTA, 1 mM EDTA, 1 mM dithiotreitol DTT, 1 mM PMSF, 10 μg/mL leupeptin, pH 7.5). Twenty micrograms of cell lysates was incubated for 1 h at 37 °C with 20 μM of the fluorogenic substrate of caspase-3 Ac-Asp-Glu-Val-Asp-7-amino-4-methylcoumarin (DEVD-AMC), in 0.25 mL of assay buffer (25 mM Hepes, 0.1% *w*/*v* 3-((3-cholamidopropyl)-dimethylammonio)-1-propanesulfonate CHAPS, 10% *w*/*v* sucrose, and 10 mM DTT, 0.01% *w*/*v* egg albumin, pH 7.5). The reaction was stopped by adding 0.75 mL of ice-cold 0.1% *w*/*v* trichloroacetic acid, and the fluorescence of AMC fragment released by active caspases was read using a Synergy HT Multi-Detection Microplate Reader (Bio-Tek Instruments, Winooski, VT, USA). Excitation and emission wavelengths were 380 and 460 nm, respectively. Fluorescence was converted in nmoles AMC/mg cellular proteins, using a calibration curve prepared previously with standard solutions of AMC.

### 2.7. Cell Viability

Cell viability was evaluated using the ATPLite kit (PerkinElmer, Waltham, MA, USA). The results were expressed as percentage of viable cells in each experimental condition versus untreated cells (considered 100% viable). 

### 2.8. Proximity Ligation Assay

The CRT–Pgp interaction was measured with the DuoLink In Situ Kit (Sigma-Merck), as per manufacturer’s instructions, using the mouse anti-Pgp (UIC-2, Millipore; dilution 1/50) and the rabbit anti-CRT (PA3-900, ABR-Affinity BioReagents Inc.; dilution 1/50) antibodies. Cell nuclei were counterstained with 4′,6-diamidino-2-phenylindole (DAPI). Cells were examined using a Leica DC100 fluorescence microscope (Leica Microsystem, Wetzlar, Germany). 

### 2.9. Confocal Microscope Analysis

Cells were seeded onto glass coverslips, and transduced with the CellLight Early Endosomes-GFP Reagent BacMam 2.0 (Invitrogen, Milan, Italy), containing an expression vector for green fluorescent protein (GFP)-Rab5a, according to the manufacturer’s instructions. Cells were then fixed using 4% *v*/*v* paraformaldehyde for 15 min, washed with PBS, and incubated for 1 h at room temperature with a red phycoerythrin (PE)-conjugated anti-Pgp (UIC-2, Millipore; dilution 1/100) antibody, washed with PBS and deionized water. Cells were examined using a Leica TCS SP2 AOP confocal laser scanning microscope. 

### 2.10. Surface CRT Expression, ATP and HMGB1 Release

To measure the levels of surface CRT, 1 × 10^5^ cells were washed with PBS, detached with Cell Dissociation Solution (Sigma-Merck), and incubated for 45 min at 4 °C with the anti-CRT antibody (Affinity Bioreagents; dilution 1/100), followed by an AlexaFluor488-conjugated secondary antibody (Abcam; dilution 1/50) for 30 min at 4 °C. After the fixation step in 2.5% *v*/*v* paraformaldehyde for 5 min at room temperature, samples were analyzed with a Guava EasyCyte (Millipore) flow cytometer equipped with the InCyte software (Millipore). Propidium iodide-negative, CRT-positive cells were counted. Cells incubated with an isotype control antibody, followed by secondary antibody, were included as control of specificity. The ATP release was measured on 100 μL of the cell culture medium with the ATP Bioluminescent Assay Kit (FL-AA, Sigma-Merck). The results were expressed as nmoles/mg cellular proteins. The release of HMGB1 was measured using the High Mobility Group Protein 1 ELISA kit (Cloud-Clone Corp., Houston, TX, USA), as per manufacturer’s instructions. The results were expressed in nmoles/mg cellular proteins.

### 2.11. Phagocytosis and T Lymphocyte Activation

DCs were generated from monocytes, immuno-magnetically isolated from peripheral blood of healthy donors, provided by Blood Bank of AOU Città della Salute e della Scienza, Torino, Italy, as previously reported [28]. The Ethical Approval by Institutional Review Board (Comitato Etico Interaziendale A.O.U. Citta` della Salute e della Scienza di Torino—A.O. Ordine Mauriziano—A.S.L. TO1) is: DG 767/2015. The phagocytosis assay was performed as detailed in [6], by co-incubating DCs and tumor cells at 37 °C and 4 °C for 24 h. The percentage of phagocytized cells obtained after the incubation at 4 °C was subtracted from the percentage obtained at 37 °C, and was always less than 5% (not shown). The phagocytosis rate was expressed as phagocytic index [6]. After cell phagocytosis, DCs were washed and co-cultured for 10 days with autologous T-cells, isolated by immuno-magnetic sorting with the Pan T Cell Isolation Kit (Miltenyi Biotec., Tetrow, Germany). The expression of CD107, a degranulation marker and an index of active CD8^+^CTLs, was determined by flow cytometry [16], using anti-human fluorescein isothyocyanate (FITC)-conjugated-CD8 (clones BW135/80 and 53–6.7; dilution 1/10) and PE-conjugated-CD107 (clones H4A3 and 1D4B; dilution 1/10) antibodies (Miltenyi Biotec.). The production of IFN-γ in the culture supernatant of CD8^+^T-cells co-cultured with DCs—a second parameter of CD8^+^T-cells cytotoxic activity [28]—was measured with the Human IFN-γ DuoSet Development Kit (R&D Systems, Minneapolis, MN, USA). Results were expressed as ng/mL.

### 2.12. Generation of Pgp-Knocked out (KO) Clones

5 × 10^5^ cells were transduced with 1 μg CRISPR pCas vectors (#KN414470, Origene, Rockville, MD, USA) targeting Pgp, respectively, or with 1 μg non-targeting vector (Origene), following the manufacturer’s instructions. Stable KO cells were selected from medium containing 1 μg/mL puromycin for 4 weeks.

### 2.13. Statistical Analysis

All data in the text and figures are provided as mean + SD. The results were analyzed by one-way ANOVA test (without corrections for multiple assessments) using Statistical Package for Social Science (SPSS) software (IBM SPSS Statistics v.19). 

## 3. Results

### 3.1. R-3 Increases Doxorubicin Intracellular Content and Cytotoxicity by Inhibiting Pgp Activity

We first analyzed the potential of *R*-3 in reversing doxorubicin resistance in human cancer cell lines of different origin (i.e., colon, lung, and breast cancer), characterized by sensitivity or acquired resistance to the drug [29]. Compared to the sensitive counterparts, HT29/DX and A549/DX cells had higher levels of Pgp and MRP1, and MDA-MB-231/DX cells had increased levels of Pgp (Figure 1A). Neither *R*-3 nor Tariquidar, used at a concentration comparable to their EC_50_ [23,24], increased doxorubicin intracellular content within Pgp-lowly expressing HT29, A549, and MDA-MB-231 cells, but they did so in Pgp-expressing variants (Figure 1B). Like Tariquidar, *R*-3 specifically reduced the catalytic activity of Pgp (Figure 1C), but it did not change the ATPase rate of other ABC transporters involved in doxorubicin efflux, such as MRP1 and BCRP (Appendix A). Notably, in MDA-MB-231/DX cells, *R*-3 increased doxorubicin’s Km and decreased Vmax of doxorubicin efflux, whereas Tariquidar only increased Km (Figure 1D; Table 1). This trend suggests that *R*-3 reduced the affinity of doxorubicin for Pgp and likely decreased the amount of active Pgp.

As expected, doxorubicin increased the rate of apoptosis, indicated by the activity of caspase 3 (Figure 2A), and reduced the viability (Figure 2B) in sensitive HT29, A549, and MDA-MB-231 cells, but it was ineffective in the resistant variants. *R*-3 increased the rate of apoptosis and decreased the cell viability upon doxorubicin treatment, as did Tariquidar (Figure 2A,B). 

To investigate if *R*-3 and Tariquidar were also able to rescue the doxorubicin-induced ICD, we next focused on MDA-MB-231 cells and their resistant counterpart, where Pgp was the main transporter involved in doxorubicin efflux.

### 3.2. R-3 but Not Tariquidar Restores Doxorubicin-Induced Immunogenic Death in Resistant Cell Lines

Doxorubicin, alone or combined with *R*-3 and Tariquidar, increased the pre-apoptotic translocation of CRT on cell surface (Figure 3A) and the release of ATP (Figure 3B) and HMGB1 (Figure 3C) in sensitive MDA-MB-231 cells. In keeping with this result, doxorubicin-treated cells were more phagocytized (Figure 3D; Appendix A); after phagocytosis, DCs induced the expansion of activated CTLs, i.e., CD8^+^CD107a^+^T-cells (Figure 3E) releasing IFN-γ (Figure 3F). Likely as a consequence of the lower intracellular accumulation (Figure 1B), doxorubicin did not elicit any immunogenic effect in the MDA-MB-231/DX variant. Unexpectedly, although Tariquidar and *R*-3 increased intracellular accumulation of doxorubicin (Figure 1B), CRT translocation (Figure 3A), and ATP (Figure 3B) and HMGB1 (Figure 3C) release occurred to a similar extent in resistant cells, and only R-3, but not Tariquidar, increased the cell phagocytosis (Figure 3D) and the percentage of activated CD8^+^T-lymphocytes (Figure 3E,F) to the same level of sensitive MDA-MB-231 cells.

### 3.3. Pgp Internalization and Degradation Restores Immunogenic Cell Death

Tariquidar is a competitive inhibitor of Pgp [30], while *R*-3 is not [23], although they both reduced the efflux of doxorubicin through Pgp. To clarify why these two inhibitors of Pgp produced such different effects on ICD, we wondered whether they affected the amount of surface Pgp that we previously identified as an inhibitor of CRT immunogenic functions [19]. 

We first investigated the interaction between Pgp and CRT on plasma membrane, using a proximity ligation approach. As expected, untreated cells or cells treated with Tariquidar or *R*-3 alone, where CRT translocation was absent, did not produce any signal of Pgp-CRT interaction (Figure 4A). Doxorubicin-treated MDA-MB-231 cells, which had constitutively amounts of Pgp (Figure 1A) and induced the translocation of CRT (Figure 3A), showed Pgp-CRT proximity, in contrast with doxorubicin-treated MDA-MB-231/DX cells that had high levels of Pgp (Figure 1A) but did not expose CRT (Figure 3A). Both doxorubicin-sensitive and doxorubicin-resistant sublines treated with Tariquidar and doxorubicin showed an interaction between Pgp and CRT (Figure 4A), suggesting that in this condition the two proteins where associated on cell surface. The presence of Pgp and its interaction with CRT may explain why Tariquidar did not induce phagocytosis in MDA-MB-231/DX cells (Figure 3D), although it increased doxorubicin content (Figure 1B) and CRT translocation (Figure 3A). By contrast, MDA-MB-231/DX cells treated with *R*-3 and doxorubicin did not display any signal of interaction (Figure 4A), notwithstanding the translocation of CRT (Figure 3A). Notably, the combination of *R*-3 and doxorubicin restored ICD in these resistant cells (Figure 3). 

We next measured the amount of Pgp and CRT on plasma membrane extracts by a biotinylation assay. MDA-MB-231 cells showed low levels of Pgp that were not affected by doxorubicin or Tariquidar; the same was true in MDA-MB-231/DX cells, although the amount of Pgp on the cell surface was higher. Interestingly, Pgp on plasma membrane was drastically reduced by *R*-3 (Figure 4B). As far as CRT is concerned, the results of biotinylation assay were in line with the flow cytometry measures (Figure 3A): doxorubicin increased surface CRT in MDA-MB-231 cells, not in the resistant variant. In both sensitive and resistant cells, the associations of doxorubicin plus Tariquidar, or doxorubicin plus *R*-3, induced the translocation of CRT to the plasma membrane. Consistently with the relative amounts of Pgp and CRT, co-immunoprecipitation assays on plasma membrane extracts revealed a weak association between Pgp and CRT in MDA-MB-231 cells upon treatment with doxorubicin or Tariquidar plus doxorubicin, and a strong association in MDA-MB-231/DX cells treated with Tariquidar plus doxorubicin. Notably, *R*-3-treated cells did not display any co-immunoprecipitation of Pgp and CRT, likely as a consequence of the reduced amount of Pgp on plasma membrane (Figure 4B). Such decrease was not due to a reduced amount of Pgp in cytosolic extracts (Appendix A).

Next, we investigated if the decrease in Pgp from the plasma membrane elicited by *R*-3 was due to its internalization. The co-labeling of MDA-MB-231/DX cells for Pgp and Rab5a, an early endosome marker, indicated that in *R*-3-treated cells Pgp was internalized within the endosomal compartment (Figure 4C). Moreover, *R*-3 induced a strong ubiquitination of Pgp, not elicited by doxorubicin nor Tariquidar (Figure 4D). This experimental set indicates that *R*-3, differently from Tariquidar, is a good inducer of ICD not only because it increased the doxorubicin content, but also because it reduced the amount of Pgp interacting with CRT on cell surface.

### 3.4. Pgp Removal is Necessary to Restore Immunogenic Cell Death in Doxorubicin-Resistant Cells

As further confirmation that Pgp removal is required to induce ICD in chemoresistant cells, we knocked out (KO) the protein in MDA-MB-231/DX cells (Figure 5A). Pgp-KO cells increased the exposure of CRT in response to doxorubicin (Figure 5B), as well as the release of ATP (Figure 5C) and HMGB1 (Figure 5D). After co-culture with DCs, KO cells treated with doxorubicin were well phagocytized (Figure 5E) and able to expand CD8^+^CD107a^+^CTLs (Figure 5F) producing IFN-γ (Figure 5G). Overall, in MDA-MB-231/DX cells treated with doxorubicin, Pgp removal restored the ICD-related parameters (Figure 5) to the same level as doxorubicin-sensitive cells (Figure 3).

## 4. Discussion

In this work, we investigated the role of Pgp in inducing resistance to ICD in cancer cells and we evaluated which types of Pgp inhibitors can be ICD-restoring agents.

To this aim, we screened different cell lines with acquired resistance to doxorubicin, consequent to a stepwise selection with the drug [25]. This strategy mimics in vitro the progressive exposure to a cumulative dosage of chemotherapy that may lead to the acquisition of chemoresistance in patients. The cell lines generated all had higher Pgp levels, decreased accumulation of doxorubicin, reduced apoptosis, and higher cell viability if treated with doxorubicin, independently from the tumor type. We previously observed that Pgp-expressing variants are resistant to ICD when treated with doxorubicin, at a dosage that elicits ICD in parental doxorubicin-sensitive counterparts [15,16]. This resistance to doxorubicin-induced ICD was observed also in MDA-MB-231/DX cells, i.e., the doxorubicin-resistant variant with the highest levels of Pgp and the lowest retention of doxorubicin, among the cell lines tested. Indeed, MDA-MB-231/DX cells did not translocate CRT, nor release ATP and HMGB1. In keeping with these findings, they were not phagocytized by DCs and were not able to expand activated (i.e., CD107a^+^, IFN-γ-producing) CTLs.

One reason for this ICD resistance could be the active efflux of doxorubicin by Pgp that limits the drug’s intracellular retention. The low amount of the drug inside the cells became insufficient to elicit ER stress [31], a key event in doxorubicin-induced ICD [2,6,7]. We thus evaluated if pharmacological inhibitors of Pgp may restore doxorubicin-induced ICD in Pgp-expressing cells.

We compared Tariquidar, the latest clinically tested Pgp inhibitor [22], and *R*-3, a N,N-bis(alkanol)amine aryl ester derivative recently synthesized by our group, with an EC_50_ for Pgp similar to Tariquidar [23,24]. Tariquidar is reported to bind to the drug binding pocket of the protein, in particular, to the so-called H-site, which is localized within the inner leaflet of the membrane at 10.5 to 14.5 Ǻ apart from the membrane surface [32], and it was suggested that it inhibits Pgp pumping from the cytoplasmic face of the membrane [33]. This compound has shown a great ability to potentiate the toxicity of doxorubicin in different cancer cells overexpressing Pgp [34,35]. *R*-3 belongs to a library of compounds that bind to the hydrophobic pocket and with transmembrane helixes of Pgp: these interactions likely alter the Pgp functions [23]. Both Tariquidar and *R*-3 indeed decreased Pgp ATPase activity and increased doxorubicin retention and toxicity. Next, we analyzed in details the effects of *R*-3 on doxorubicin efflux kinetics: we found that Tariquidar increased Km but did not change Vmax, whereas *R*-3 produced higher Km and lower Vmax. The altered Km indicates a lower affinity of doxorubicin for Pgp, suggesting that the compounds interfere with the doxorubicin binding to the transporter. The lower Vmax is instead indicative of a lower amount of active Pgp present on the cell surface [26]. Notably, both inhibitors induced the exposure of CRT on cell surface of MDA-MB-231/DX cells treated with doxorubicin, as well as the release of ATP and HMGB1. These two actions suggested that they both release the tumor-dependent signals necessary to trigger ICD. Unexpectedly, however, Tariquidar-treated cells were not phagocytized by DCs nor were able to activate CD8^+^CTLs, while *R*-3-treated cells underwent a complete ICD process.

Besides the endogenous signals of cancer cells, ICD is complete when DCs are attracted within the tumor bed, primed to phagocytize, and mature. ATP is a soluble inducer of all these processes [36]. Furthermore, DCs are fully engaged by direct ligand–receptor contacts with tumor cells, such as the contacts between CRT/ERp57 and LDL receptor-related protein1/CD91 [6,7] or between HSP90 and TLR4 [11]. Pgp inhibits the CRT-mediated phagocytosis of cancer cells, likely by a direct interaction with CRT in colon cancer HT29/DX cells [19]. The proximity ligation and co-immunoprecipitation assays of surface proteins confirmed a physical interaction between Pgp and CRT in sensitive MDA-MB-231 cells treated with doxorubicin, alone or with Tariquidar, i.e., two conditions associated with increased intracellular doxorubicin and increased surface CRT. As expected, such interaction was not detectable in resistant MDA-MB-231/DX cells treated with doxorubicin that was poorly accumulated within cells. CRT–Pgp interaction was detected in cells exposed to doxorubicin plus Tariquidar, characterized by an increased doxorubicin content and CRT translocation. By contrast, both sensitive and resistant cells treated with *R*-3 had no signals of interaction between Pgp and CRT, despite a translocation of CRT comparable to Tariquidar-treated cells. The lack of signal in proximity ligation and co-immunoprecipitation assays was due to the strong reduction of Pgp on the cell surface, elicited by *R*-3. This data explains the lower Vmax observed in doxorubicin efflux. Moreover, it explains why *R*-3-treated cells—with a reduced amount of Pgp on their surface—were well phagocytized by DCs.

CRT has distinct domains important for its correct folding and immunogenic functions: the highly conserved amino-terminal domain and the close P domain constitute most of the extracellular portion of CRT interacting with other proteins [37]. On the other hand, Pgp is an integral membrane protein, characterized by 12 transmembrane helixes, connected by multiple extracellular loops and two large intracellular nucleotide binding domains [21]. The shift from an inward open conformation to an outward open conformation upon drug binding and efflux [21] makes Pgp a highly dynamic protein, able to transiently interact with other ecto-enzymes. Such interactions can modulate Pgp activity, as in the case of Pgp-carbonic anhydrase XII interaction [25], or modulate the function of the client protein. This is the case in the interaction between CRT and Pgp [19], where we hypothesize that Pgp masks specific CRT domains critical for its immunogenic activity. We are currently producing CRT and Pgp mutants to clarify the possible sites of interaction between the two proteins.

Lowering the amount of Pgp from cell surface, as in the case of *R*-3-treated cells, indeed relieved the inhibition on CRT, allowing the phagocytosis of cancer cells and the subsequent expansion of activated CD8^+^CTLs. In particular, *R*-3 reduced the amount of surface Pgp by inducing its internalization via endosome and subsequent ubiquitination. Pgp internalization can follow either endosomal or lysosomal pathways. The endosomal pathway, supported by the observation that Pgp is associated with the early endosome marker Rab5a [38,39], reduces the amount of Pgp from cell plasma membrane and results in increased accumulation of Pgp substrates [40]. By contrast, when Pgp is internalized via lysosomes, it maintains a high activity of Pgp on the lysosome membrane: this type of endocytosis induces chemoresistance, because it sequesters the chemotherapeutic drugs within lysosomes [41]. The endosome-associated internalization, but not the lysosomal one, results in Pgp ubiquitination [40]. Until now, few agents are described as inducers of Pgp ubiquitination. Among them, there are sulfhydrating agents such as hydrogen disulfide-releasing doxorubicin [42] or disulfiram [43] that destabilize disulfide bonds and alter the proper folding of nascent Pgp. Curiously, the tetrandrine derivative H1, an inhibitor of Pgp ATPase activity, has been reported to be a strong inducer of Pgp ubiquitination [44]. We suggest that *R*-3, interacting with the drug binding pocket and interfering with the Pgp ATPase cycle, also destabilized the protein and primed it for ubiquitination. 

The proof of concept that a high expression of Pgp inhibits ICD was provided by Pgp-KO MDA-MB-231/DX cells. In these cells, the increase in CRT exposure induced by doxorubicin was followed by a strong DC-mediated phagocytosis and an increase in CD8^+^CD107a^+^/IFN-γ-producing CTLs. This experimental set indicates that doxorubicin-resistant cells reacquired a complete sensitivity to ICD, behaving like parental drug-sensitive cells, if Pgp is removed from cell surface. 

Triple-negative breast cancer, i.e., the cell model evaluated in the present work, is of particular interest because it is treated with anthracycline-based chemotherapy. Unfortunately, it is often chemoresistant because of the high expression of Pgp [43], and therefore characterized by poor prognosis [45]. On the other hand, breast cancer patients with high CD8^+^CTL infiltration, which correlated with a higher intratumor expression of CRT, have a better survival [46]. Increasing the ICD of triple-negative breast cancer cells with compounds other than conventional chemotherapeutic drugs [47] has been explored in this resistant cancer type. Our data indicate that specific Pgp inhibitors may be useful for this purpose. As most of the Pgp inhibitors failed as chemosensitizers agents [20], their use as ICD-inducers in combination with doxorubicin appears as an interesting repurposing option. 

## 5. Conclusions

In this work, we demonstrated that doxorubicin-resistant/Pgp-expressing cancer cells are also resistant to the doxorubicin-induced ICD for at least two reasons: the lower accumulation of doxorubicin within the cells, due to the active drug efflux via Pgp, and the presence of Pgp itself that impairs the ability of CRT to recruit DCs and stimulate phagocytosis. Disrupting the CRT-Pgp interaction using pharmacological Pgp inhibitors may re-sensitize ICD-resistant cells to the immunogenic effect of chemotherapy. However, inhibitors reducing only the efflux activity of Pgp restore doxorubicin cytotoxic effects but not ICD. By contrast, inhibitors reducing the activity and amount of Pgp are effective chemo-immunesensitizing compounds. We suggest that this type of pharmacological inhibitors may be useful in chemo-immunetherapeutic protocols for resistant tumors.

## Figures and Tables

**Figure 1 cells-09-01033-f001:**
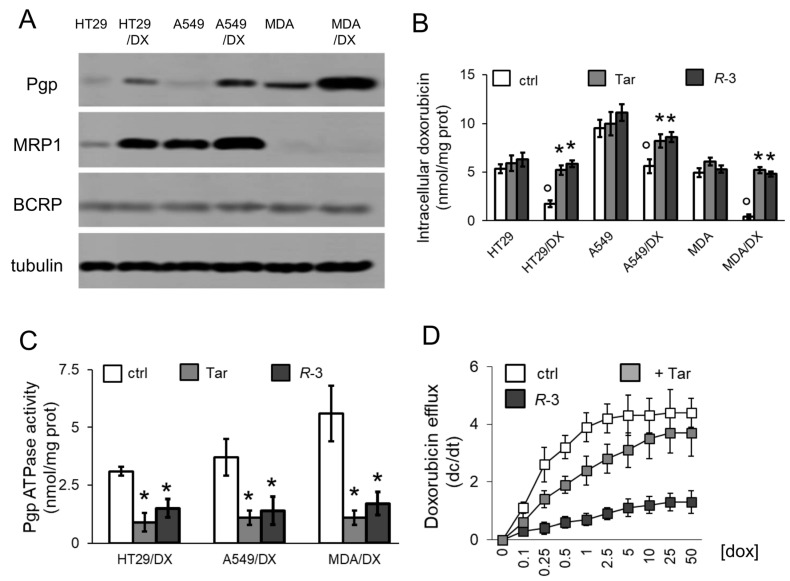
*R*-3 increases doxorubicin intracellular content by inhibiting Pgp activity. (**a**) Immunoblotting of the indicated proteins in human colon cancer HT29 cells, non-small cell lung cancer A549 cells, triple-negative breast cancer MDA-MB-231 cells, and in their doxorubicin (DX)-resistant variants. Tubulin was used as control of equal protein loading. The image is representative of 1 out of 3 experiments. (**b**) Cells were incubated for 3 h in medium containing 5 μM doxorubicin (dox), in the absence (ctrl) or presence of 75 nM Tariquidar (Tar) or 75 nM *R*-3. Intracellular doxorubicin content was measured fluorimetrically in duplicates (*n* = 3). Data are means ± SD. * *p* < 0.005: treated cells vs. respective ctrl cells; ° *p* < 0.001: DX cells vs. parental cells. (**c**) Cells were grown for 24 h in the absence (ctrl) or presence of 75 nM Tariquidar (Tar) or 75 nM *R*-3. Pgp ATPase activity was measured spectrophotometrically on membrane extracts of DX variants. Data are means ± SD (*n* = 3). * *p* < 0.001: treated cells vs. respective ctrl cells. (**d**) MDA-MB-231/DX cells were grown in the absence (ctrl) or presence of 75 nM Tariquidar (Tar) or 75 nM *R*-3 for 24 h, then incubated for 20 min with increasing concentrations (0–50 μM) of doxorubicin (dox). The procedure was repeated on a second series of dishes, incubated in the same experimental conditions and analyzed after 10 min. Data are means ± SD (*n* = 3). The rate of doxorubicin efflux (dc/dt) was plotted versus the intracellular concentration of the drug.

**Figure 2 cells-09-01033-f002:**
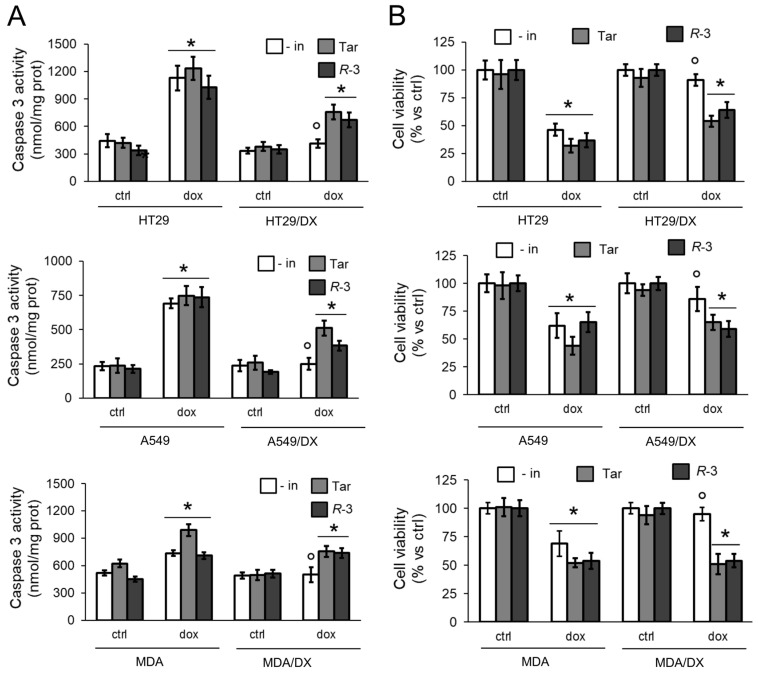
*R*-3 increases doxorubicin-induced apoptosis and cytotoxicity in Pgp-expressing cells. (**a**) Human colon cancer HT29 cells, non-small cell lung cancer A549 cells, triple-negative breast cancer MDA-MB-231 cells, and their doxorubicin (DX)-resistant variants were incubated 24 h (**a**) or 48 h (**b**), in fresh medium (ctrl), without inhibitors (- in) or with 75 nM Tariquidar (Tar) or 75 nM *R*-3. When indicated, 5 μM doxorubicin (dox) was added. (**a**) Activity of caspase 3 was measured fluorimetrically in duplicates (*n* = 3). Data are means ± SD. * *p* < 0.001: treated cells vs. respective ctrl cells; ° *p* < 0.001: DX cells vs. parental cells. (**b**) Cell viability was measured by a chemiluminescence-based assay in quadruplicates. Data are means ± SD (*n* = 3). * *p* < 0.001: treated cells vs. respective ctrl cells; ° *p* < 0.05: DX cells vs. parental cells.

**Figure 3 cells-09-01033-f003:**
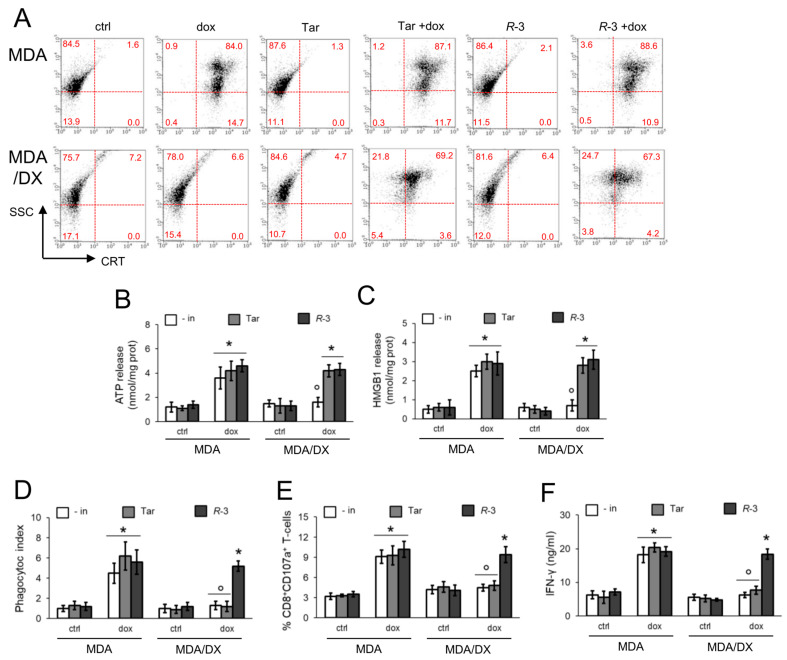
*R*-3 but not Tariquidar rescues the immunogenic cell death induced by doxorubicin in Pgp-expressing cells. (**a**) Human triple-negative breast cancer MDA-MB-231 cells and their doxorubicin (DX)-resistant variant were incubated 6 h (**a** and **b**) or 24 h (**c**–**f**), in fresh medium (ctrl) or in medium containing 5 μM doxorubicin (dox). Cells were grown without inhibitors (- in), with 75 nM Tariquidar (Tar) or 75 nM *R*-3. (**a**) Dot plots representative of surface calreticulin (CRT), performed by flow cytometry in duplicate (*n* = 3). Numbers represent the percentage of cells/quadrant. Cut-off for positivity was fixed at 10^2^. SSC: side-scatter. The image is representative of 1 out of 3 experiments. (**b**) Release of extracellular ATP, measured by a chemiluminescence-based assay in duplicates (*n* = 3). Data are means ± SD. * *p* < 0.001: treated cells vs. respective ctrl cells; ° *p* < 0.001: DX cells vs. parental cells. (**c**) Release of extracellular HMGB1, measured by ELISA in duplicates (*n* = 3). Data are means ± SD. * *p* < 0.001: treated cells vs. respective ctrl cells; ° *p* < 0.001: DX cells vs. parental cells. (**d**) After the treatments indicated in panel (**c**), tumor cells were stained with FITC-PKH2 dye and DCs were stained with an anti-PE-HLA-DR antibody. Tumor cells were co-incubated with DCs for 24 h. Double-stained cells were counted by flow cytometry. Data are means ± SD (*n* = 3). * *p* < 0.001: treated cells vs. respective ctrl cells; ° *p* < 0.001: DX cells vs. parental cells. (**e**) T-lymphocytes were co-cultured with DCs after phagocytosis, and then incubated 24 h with MDA-MB-231 or MDA-MB-231/DX cells. The percentage of CD8^+^CD107a^+^T-cells was measured by flow cytometry. Data are means ± SD (*n* = 3). * *p* < 0.001: treated cells vs. respective ctrl cells; ° *p* < 0.001: DX cells vs. parental cells. (**f**) IFN-γ concentration was measured in the supernatants of T-lymphocytes, treated as indicated in (**e**), by ELISA in duplicates. Data are means ± SD (*n* = 3). * *p* < 0.001: treated cells vs. respective ctrl cells; ° *p* < 0.001: DX cells vs. parental cells.

**Figure 4 cells-09-01033-f004:**
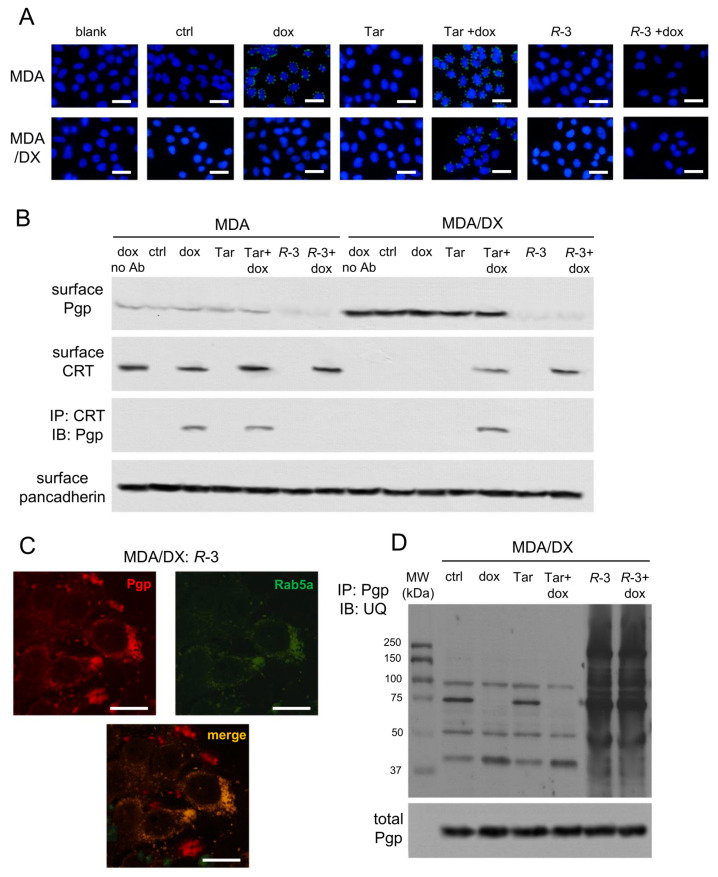
*R*-3 disrupts Pgp-calreticulin interaction, favoring the internalization and ubiquitination of Pgp. Human triple-negative breast cancer MDA-MB-231 cells and their doxorubicin (DX)-resistant variant were incubated for 24 h in fresh medium (ctrl) or in medium containing 5 μM doxorubicin (dox). When indicated, 75 nM Tariquidar (Tar) or 75 nM *R*-3 were added. (**a**) Proximity ligation assay between Pgp and calreticulin. Blank: cells incubated without primary antibodies. Blue signal: nuclear staining (DAPI). Green signal: Pgp/calreticulin interaction. The image is representative of 1 out of 3 experiments. A minimum of 5 fields/experiment was examined. Bar: 10 μm (10× ocular lens; 63× objective lens). (**b**) Plasma membrane extracts were immunoblotted and decorated to detect Pgp or CRT, or immuno-precipitated (IP) with an anti-CRT antibody, then immunoblotted (IB) with an anti-Pgp antibody. no Ab: MDA-MB-231 and MDA-MB-231/DX samples treated with doxorubicin, immuno-precipitated without antibodies. An aliquot of the extracts before the immunoprecipitation was probed with an anti-pancadherin antibody, as control of equal protein loading. The figure is representative of 1 out of 3 experiments. (**c**) MDA-MB-231/DX cells were incubated 24 h with 75 nM *R*-3. During the last 18 h, cells were transfected with an expression vector for GFP-conjugated Rab5a, to label early endosomes. Subsequently, cells were labeled with an anti-PE-Pgp antibody. Immunofluorescence detection of Pgp co-localized with Rab5a was achieved by confocal microscope analysis. The image is representative of 1 out of 3 experiments. A minimum of 5 fields/experiment was examined. Bar: 10 μm (10× ocular lens; 60× objective lens). (**d**) Whole cell extracts of MDA-MB-231/DX cells were immunoprecipitated with an anti-Pgp antibody, then immunoblotted for ubiquitin (UQ). An aliquot of the extracts before the immunoprecipitation was probed with an anti-Pgp antibody, as index of the total amount of Pgp immunoprecipitated in each sample. The figure is representative of 1 out of 3 experiments.

**Figure 5 cells-09-01033-f005:**
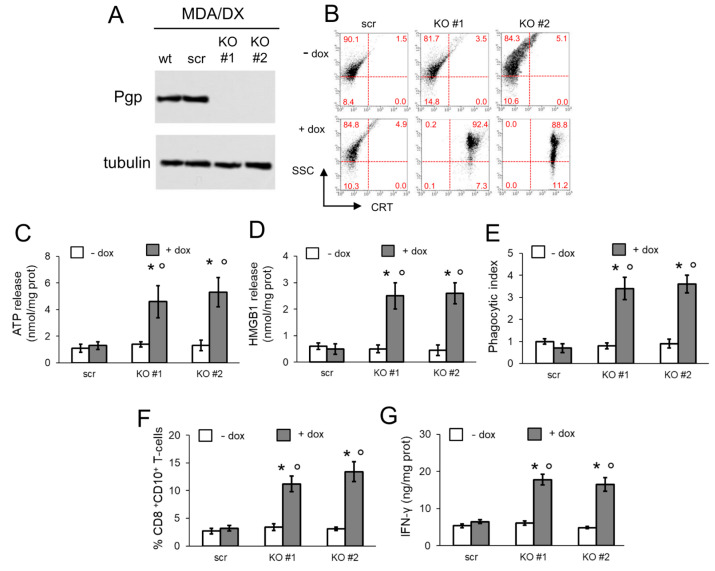
Pgp knock-out restores doxorubicin-induced immunogenic cell death. MDA-MB-231/DX cells (wild type, wt) were transduced with a non-targeting vector (scrambled vector; scr) or with two CRISPR pCas Pgp-targeting vectors (KO#1, KO#2). Cells were grown in fresh medium (- dox) or in the presence of 5 μM doxorubicin (+ dox), for 6 h (**b** and **c**) or 24 h (**d–f**). (**a**) Pgp immunoblotting in whole cell extracts. Tubulin was used as control of equal protein loading. The image is representative of 1 out of 3 experiments. (**b**) Dot plots representative of surface calreticulin (CRT), performed by flow cytometry in duplicates (*n* = 3). Numbers represent the percentage of cells/quadrants. Cut-off for positivity was fixed at 10^2^. SSC: side-scatter. The image is representative of 1 out of 3 experiments. (**c**) Release of extracellular ATP, measured by a chemiluminescence-based assay in duplicates (*n* = 3). Data are means ± SD. * *p* < 0.001: KO-cells vs. scr-cells; ° *p* < 0.001: + dox cells vs. - dox cells. (**d**) Release of extracellular HMGB1, measured by ELISA in duplicates (*n* = 3). Data are means ± SD. * *p* < 0.001: KO-cells vs. scr-cells; ° *p* < 0.001: + dox cells vs. - dox cells. (**e**) After the treatment indicated in panel (**d**), tumor cells were stained with FITC-PKH2 dye, DCs were stained with an anti-PE-HLA-DR antibody. Tumor cells were co-incubated with DCs for 24 h. Double-stained cells were counted by flow cytometry. Data are means ± SD (*n* = 3). * *p* < 0.001: KO-cells vs. scr-cells; ° *p* < 0.001: + dox cells vs. - dox cells. (**f**) T-lymphocytes were co-cultured with DCs after phagocytosis, and then incubated 24 h with scr- or KO-cells. The percentage of CD8^+^CD107a^+^T-cells was measured by flow cytometry. Data are means ± SD (*n* = 3). * *p* < 0.001: KO-cells vs. scr-cells; ° *p* < 0.001: + dox cells vs. - dox cells. (**g**) IFN-γ concentration was measured in the supernatants of T-lymphocytes, treated as indicated in panel (f), by ELISA in duplicates. Data are means ± SD (*n* = 3). * *p* < 0.001: KO-cells vs. scr-cells; ° *p* < 0.001: + dox cells vs. - dox cells.

**Table 1 cells-09-01033-t001:** Kinetic parameters of doxorubicin efflux in MDA-MB-231/DX cells.

Experimental Condition	Km	Vmax
ctrl	0.19 + 0.04	4.41 + 0.56
+ Tar	0.94 + 0.15 *	3.82 + 0.83
+ *R*-3	0.76 + 0.19 *	1.32 + 0.42 *^,1^

^1^ MDA-MB-231/DX cells were grown in the absence (ctrl) or presence of 75 nM Tariquidar (Tar) or 75 nM *R*-3 for 24 h, and then treated as reported in Figure 1D. Km (μM) and Vmax (μmoles/min) were calculated with the Enzfitter software. * *p* < 0.001: treated cells vs. respective ctrl cells.

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
