# Peer review of "Insights into P-Glycoprotein Inhibitors: New Inducers of Immunogenic Cell Death"

_cells, 2020, doi:10.3390/cells9041033_

Round 1

Reviewer 1 Report

Authors found a N,N-bis(alkanol)amine aryl ester derivative (R3) of Tariquidar, promoted phagocytosis of doxorubicin-treated tumour cells by dendritic cells and activation of anti-tumor CD8+T-lymphocytes as well as decreased doxorubicin efflux and resistance in cancer cells. Unlike tariquidar, R-3 promoted Pgp internalization and ubiquitination, disrupting its interaction with calreticulin (CRT), which is potent attractive signals to engage dendritic cells and expand CD8+ cytotoxic T lymphocytes. While I appreciate the novel and interesting results, some questions should be addressed before acceptance for publication in Cells.

  1. Is the Pgp ATPases activity determined as the vanadate-sensitive inorganic phosphate production rate?
  2. It would be necessary to show the representative raw data of DC-mediated phagocytosis of doxorubicin-treated tumour cells in the presence and absence of R3, even in the supplementary data file.

Some minors as follows.

Line 27-28: Should change into “Pgp knock-out restores doxorubicin-induced immunogenic cell death in MDA-MB-231/DX cells”

Line 31: “Ampunt” should be “amount”

Author Response

Authors found a N,N-bis(alkanol)amine aryl ester derivative (R3) of Tariquidar, promoted phagocytosis of doxorubicin-treated tumour cells by dendritic cells and activation of anti-tumor CD8+T-lymphocytes as well as decreased doxorubicin efflux and resistance in cancer cells. Unlike tariquidar, R-3 promoted Pgp internalization and ubiquitination, disrupting its interaction with calreticulin (CRT), which is potent attractive signals to engage dendritic cells and expand CD8+ cytotoxic T lymphocytes. While I appreciate the novel and interesting results, some questions should be addressed before acceptance for publication in Cells.

  1. Is the Pgp ATPases activity determined as the vanadate-sensitive inorganic phosphate production rate?

The Pgp ATPase activity was the vanadate-sensitive phosphate hydrolysis. We specified this point in the Materials and Methods (line 137).

  1. It would be necessary to show the representative raw data of DC-mediated phagocytosis of doxorubicin-treated tumour cells in the presence and absence of R3, even in the supplementary data file.

We added the raw data as new Supplementary Figure S3. We modified text (line 263) and Supplementary Materials accordingly.

Some minors as follows.

- Line 27-28: Should change into “Pgp knock-out restores doxorubicin-induced immunogenic cell death in MDA-MB-231/DX cells”

- Line 31: “Ampunt” should be “amount”

We modified the sentences indicated as minor points.

Reviewer 2 Report

This manuscript presents results from in vitro experiments that evaluate the effects of P-glycoprotein inhibitors on an immunogenic cell death (ICD) pathway mediated by doxorubicin (dox). The authors found that an approved inhibitor (Tariquidar) does not induce ICD in dox-resistant cells, while inhibitor compound R-3 does inhibit ICD through downregulating P-glycoprotein in dox-resistant cells. While the studies are similar to others presented in the literature from the authors and from other investigators, this manuscript presents new results with compound R-3 and how this compound appears to restore ICD through P-glycoprotein-mediated effects, and potential interactions between P-glycoprotein and calreticulin that may influence ICD. The methods employed are described adequately and are appropriate for the studies. The results are presented clearly. The conclusions appear to be justified based on the results presented. This reviewer recommends the following points should be addressed by the authors:

Comments to be addressed:

-Figure S2 caption states that both parental (dox-sensitive) and dox-resistant cells were used, but the figure does not show data for parental cells; please revise the figure caption to match what is presented in the figure.

-Please explain why Fig. 4B has surface CRT present in the “dox/noAb” lane of the blot from DX (resistant) cells (the presence of a significant amount of CRT here seems to be inconsistent with the flow cytometry data presented in Fig. 3A).

-This reviewer recommends changing the title of the manuscript; while data does suggest interactions between calreticulin and P-glycoprotein, the interaction is not well characterized in the studies presented in the manuscript, and more emphasis and discussion is present for other aspects of the studies—specifically, the compound R-3.

-Please indicate if the statistical comparisons between two groups included corrections for multiple assessments following one-way ANOVA.

Specific comments and recommended corrections:

- Do not include a hyphen when writing “immune system” or “immune cell” (see line 36, 39 and elsewhere) or “plasma membrane” (line 46, 118, 125, 460 and elsewhere)

-line 31: change “ampunt” to “amount”

-line 39: recommend the following revision for “make cancer cells recognized…”: “cause cancer cells to be recognized…”

-lines 42-45, the sentence needs revision (grammar)

-line 73: change “included” to “including”

-lines 78-79: some revision (especially “poorly” change to “poor”, also “high side-toxicity”)

-line 178: revise the wording of the following to increase clarity: “not-immune isotype antibody”

-lines 212-213: change to “Neither R-3 nor Tariquidar…”

-line 255: remove “s” in “Data are s mean…””

-lines 415-416: the following needs revision: “When next deepened the effects of R-3 on doxorubicin efflux kinetics: we found that Tariquidar…”

-line 461: please re-word the following to increase clarity and grammar: “…maintains high the activity of Pgp, concentrated on the lysosome membrane…”

-Most of the Conclusions section (lines 480-488, 491-492) should be moved to the Discussion, since the text is discussion of literature rather than conclusions from the studies presented in the manuscript.

Author Response

This manuscript presents results from in vitro experiments that evaluate the effects of P-glycoprotein inhibitors on an immunogenic cell death (ICD) pathway mediated by doxorubicin (dox). The authors found that an approved inhibitor (Tariquidar) does not induce ICD in dox-resistant cells, while inhibitor compound R-3 does inhibit ICD through downregulating P-glycoprotein in dox-resistant cells. While the studies are similar to others presented in the literature from the authors and from other investigators, this manuscript presents new results with compound R-3 and how this compound appears to restore ICD through P-glycoprotein-mediated effects, and potential interactions between P-glycoprotein and calreticulin that may influence ICD. The methods employed are described adequately and are appropriate for the studies. The results are presented clearly. The conclusions appear to be justified based on the results presented. This reviewer recommends the following points should be addressed by the authors:

Comments to be addressed:

-Figure S2 caption states that both parental (dox-sensitive) and dox-resistant cells were used, but the figure does not show data for parental cells; please revise the figure caption to match what is presented in the figure.

We apologize for the mistake: we modified the legend of Figure S2 accordingly.

-Please explain why Fig. 4B has surface CRT present in the “dox/noAb” lane of the blot from DX (resistant) cells (the presence of a significant amount of CRT here seems to be inconsistent with the flow cytometry data presented in Fig. 3A).

We recognize that the experiment showed in the image is likely affected by a technical bias: we hypothesize an overflow of the material from the lane before, because it is unlikely that the biotinylation assay – although very sensitive – produced a so strong signal from MDA-MB-231/DX cells treated with doxorubicin, where only 6.6% of cells were positive to surface CRT (Figure 3A). We agree that leaving this image is misleading. We preferred to substitute it with the blots of a second experiment from the same experimental set. We apologize for the inconvenience.

-This reviewer recommends changing the title of the manuscript; while data does suggest interactions between calreticulin and P-glycoprotein, the interaction is not well characterized in the studies presented in the manuscript, and more emphasis and discussion is present for other aspects of the studies—specifically, the compound R-3.

We thanks the Reviewer for the suggestion. We changed the title of the manuscript emphasizing the characterization of Pgp inhibitors as inducers of immunogenic cell death.

-Please indicate if the statistical comparisons between two groups included corrections for multiple assessments following one-way ANOVA.

We did not perform the corrections for multiple assessments. We added the statement in Materials and Methods (line 205).

Specific comments and recommended corrections:

- Do not include a hyphen when writing “immune system” or “immune cell” (see line 36, 39 and elsewhere) or “plasma membrane” (line 46, 118, 125, 460 and elsewhere)

We modified the words throughout the manuscript.

-line 31: change “ampunt” to “amount”

-line 39: recommend the following revision for “make cancer cells recognized…”: “cause cancer cells to be recognized…”

-lines 42-45, the sentence needs revision (grammar)

-line 73: change “included” to “including”

-lines 78-79: some revision (especially “poorly” change to “poor”, also “high side-toxicity”)

-line 178: revise the wording of the following to increase clarity: “not-immune isotype antibody”

-lines 212-213: change to “Neither R-3 nor Tariquidar…”

-line 255: remove “s” in “Data are s mean…””

We corrected all the typos and sentences indicated above.

-lines 415-416: the following needs revision: “When next deepened the effects of R-3 on doxorubicin efflux kinetics: we found that Tariquidar…”

We rephrased the sentence.

-line 461: please re-word the following to increase clarity and grammar: “…maintains high the activity of Pgp, concentrated on the lysosome membrane…”

We rephrased the sentence.

-Most of the Conclusions section (lines 480-488, 491-492) should be moved to the Discussion, since the text is discussion of literature rather than conclusions from the studies presented in the manuscript.

We thanks the Reviewer for this consideration. We moved the sentences indicated and the references in the Discussion section (line 475).